# Diclofenac Sensitizes Signet Ring Cell Gastric Carcinoma Cells to Cisplatin by Activating Autophagy and Inhibition of Survival Signal Pathways

**DOI:** 10.3390/ijms232012066

**Published:** 2022-10-11

**Authors:** Nang Lae Lae Phoo, Amonnat Sukhamwang, Pornngarm Dejkriengkraikul, Supachai Yodkeeree

**Affiliations:** 1Department of Biochemistry, Faculty of Medicine, Chiang Mai University, Chiang Mai 50200, Thailand; 2Center for Research and Development of Natural Products for Health, Chiang Mai University, Chiang Mai 50200, Thailand; 3Anticarcinogenesis and Apoptosis Research Cluster, Faculty of Medicine, Chiang Mai University, Chiang Mai 50200, Thailand

**Keywords:** gastric cancer, cisplatin resistance, diclofenac, cisplatin, autophagy, ROS, survival proteins

## Abstract

Gastric cancer has one of the highest incidence rates of cancer worldwide while also contributing to increased drug resistance among patients in clinical practice. Herein, we have investigated the role of diclofenac (DCF) on sensitizing cisplatin resistance in signet ring cell gastric carcinoma cells (SRCGC). Non-toxic concentrations of DCF significantly augmented cisplatin-induced cell death in cisplatin-resistant SRCGC cells (KATO/DDP) but not in cisplatin-sensitive SRCGC cells (KATOIII). Consistently, concomitant treatment of DCF and cisplatin significantly enhanced autophagic cell death due to overproduction of intracellular reactive oxygen species (ROS). At the molecular level, the induction of ROS has been associated with a reduction in antioxidant enzymes expression while inhibiting nuclear factor erythroid 2-related factor 2 (Nrf2) activity. Moreover, the combination of DCF and cisplatin also inhibited the expression of survival proteins including Bcl-2, Bcl-xL, cIAP1 and cyclin D1 in KATO/DDP cells when compared with cisplatin alone. This was due, at least in part, to reduce MAPKs, Akt, NF-κB, AP-1 and STAT-3 activation. Taken together, our results suggested that DCF potentiated the anticancer effect of cisplatin in SRCGC via the regeneration of intracellular ROS, which in turn promoted cell death as an autophagy mechanism and potentially modulated the cell survival signal transduction pathway.

## 1. Introduction

Gastric cancer is recognized as one of the most life-threatening malignancies worldwide, particularly in the East Asian region. *Helicobacter pylori (H. pylori)* infection has been well known as one of the strongest risk factors for the development of gastric cancer. Due to advances in the treatment of *H. pylori*, there has been a decline in incidences of gastric cancer over the past few years. However, independent strains of *H. pylori* continue to persist, and recent studies have reported an increase in incidences of signet ring cell gastric carcinoma cells (SRCGC). SRCGC is a distinct type of gastric cancer and presently it accounts for 16.8% of new gastric adenocarcinoma cases [1]. To this point, platinum combination chemotherapy is considered the gold standard in the treatment and management of signet ring cells containing gastric carcinoma [2]. However, the clinical outcomes were far from satisfactory due to the development of drug resistance.

Cisplatin resistance among gastric cancer patients can be due to combination of multiple factors and mechanisms including increased drug efflux, decreased drug uptake, alterations to the intracellular signaling pathway, enhanced DNA damage repair capacity, increased inactivation of certain drugs and a significant degree of dysregulation during apoptosis [3]. Recently, increases in antioxidant response via the enhancement of nuclear factor erythroid 2-related factor 2 (Nrf2) activity has been associated with clinically relevant cisplatin resistance in tumor cells. Nrf2 has been considered the master regulator of antioxidant and cytoprotective proteins including superoxide dismutase 1 (SOD1), thioredoxin reductase, aldoketoreductases (AKRs) and heme oxygenase-1 (HO-1) [4]. High levels of glutathione, SOD1 and HO-1 have been found to assist in preserving cancer cells in incidences of platinum drug toxicity [5]. Our previous study has determined that the inhibition of AKR1C1 and 1C3 activity could then effectively sensitized cisplatin-resistant SRCGC cells to cisplatin [6]. Therefore, it would be highly beneficial to identify a suitable drug sensitizer that can reverse chemotherapy resistance and enhance sensitivity to platinum chemotheraputic drugs via modulation redox homeostasis. A previous study reported that using non-steroidal anti-inflammatory drug (NSAID) in combination with chemotherapeutic drugs could effectively potentiate the anticancer activity of various chemotherapeutic drugs [7]. Diclofenac (DCF) is one of the most prescribed NSAID drugs in the treatment and management of inflammation. Experimental studies have indicated that DCF could induce cancer cell death. Moreover, DCF can inhibit melanoma cell growth by increasing intracellular ROS and inhibiting the activity of SOD1 [8]. In Hela and pancreatic cancer cell lines, DCF in combination with 5-flurouracil could induce the cancer cell death by impairing the autophagy flux and it pointed out that the effect of autophagy may either as cytoprotective or cytotoxic role depending on the type of chemotherapeutic agent and nature of the cell itself [9,10]. In addition, DCF effectively increased the cytotoxicity of cisplatin in combination treatments that involve the cisplatin-resistant A549 cell line [11]. Moreover, the role of DCF in inhibiting AKRs activity has been reported [12]. In this study, we aimed to explore the effects and molecular mechanism of DCF on cisplatin-induced cytotoxicity in cisplatin-resistant SRCGC cells.

## 2. Results

### 2.1. DCF Potentiated the Cytotoxicity of Cisplatin in Cisplatin-Resistant Signet Ring Cell Gastric Carcinoma

KATO/DDP cisplatin-resistant SRCGC was derived from the parental KATOIII cell line according to the method described in our previously published research work [6]. To examine the resistance ratio between KATO/DDP and KATOIII cells, both cells were treated with cisplatin 0–100 µM, and IC_50_ of each cell line was calculated. The resistance ratio of KATO/DDP cells was 7.14 times more resistant than KATOIII cells, as is shown in Figure 1A. Next, the cytotoxicity of DCF on KATOIII and KATO/DDP cells was determined by trypan blue cell exclusion assay. The results revealed that treatment of KATOIII and KATO/DDP cells with DCF below 125 μM had no effect on cell viability (>80% cell survival) as is shown in Figure 1B. To determine whether DCF could sensitize cisplatin-induced drug resistance, KATOIII and KATO/DDP cells were treated with cisplatin at 0–100 µM and in combination with DCF 10 µM and 20 µM. As is shown in Figure 1C, the cotreatment of cisplatin with DCF had no significant cytotoxic effect on KATOIII cells when compared to treatments with cisplatin alone. Conversely, the survival rate of KATO/DDP cells was reduced significantly from 74% to 61% and 45% when cisplatin at 25 µM was combined with DCF at 10 µM and 20 µM, respectively, when compared with cisplatin alone, as is presented in Figure 1D. In addition, combination treatment of cisplatin (0–50 μM) and DCF at 10 and 20 μM had no significant effect on human dermal fibroblast cell viability when compared with cisplatin alone, as shown in Figure 1E. This result indicated that DCF could enhance the cytotoxicity of cisplatin in cisplatin-resistant SRCGC.

### 2.2. DCF Enhanced the Cytotoxicity of Cisplatin in KATO/DDP Cells via Autophagy Cell Death

Next, we examined whether the sensitization effect of DCF on cisplatin-induced KATO/DDP cell death was associated with apoptosis by employing the annexin V- propidium iodide (PI) staining assay. The results indicated that treatment of the cells with cisplatin alone at 25 μM could induce the apoptotic population from 11% to 17% when compared with the control. However, a combination treatment of cisplatin with DCF at 10 and 20 µM revealed no significant increase in the apoptotic population when compared with cisplatin alone (Figure 2A,B). To confirm whether or not apoptosis is the main cause of DCF-enhanced cisplatin-induced death, the level of cleavage of PARP, as a hallmark of apoptosis activation, was determined by Western blot analysis. As is shown in Figure 2C, cotreatment with cisplatin and DCF at 10 and 20 μM exhibited no significant changes in the level of cleaved PARP when compared with cisplatin alone. These outcomes indicated that apoptotic cell death was not the major pathway of DCF-enhanced cytotoxicity of cisplatin in KATO/DDP cells.

Cisplatin exerts a cytotoxic effect not only by inducing apoptosis, but also by autophagic cell death. A key objective of this study was to determine whether DCF could enhance cisplatin-induced cell death via autophagy. Accordingly, KATO/DDP cells were treated with 25 μM of cisplatin combined with 10 and 20 μM of DCF. The autophagic vacuoles were then stained with monodansyl cadaverine (MDC) labelled dye. As is shown in Figure 2D,E, the formation of autophagic vacuoles in KATO/DDP cells increased from 27% to 39% and 45% in cotreatments involving cisplatin and DCF at 10 and 20 μM, respectively, when compared with treatment of cisplatin alone. Other than increasing the formation of autophagy vacuoles, an increase was also observed in the formation of light chain 3B-II (LC3B-II), the autophagy marker, after administering a combination treatment of 25 μM cisplatin and DCF at 10 and 20 μM (Figure 2F). These results indicated that DCF enhanced cisplatin cytotoxicity by activating autophagy cell death. Next, we confirmed that DCF enhanced the cytotoxicity of cisplatin via the autophagy pathway by using 3-methyladenine (3MA), an autophagy inhibitor. As is shown in Figure 2G, the survival of KATO/DDP cells was significantly increased from 60% to 75% in cells treated with 3MA along with cisplatin 25 μM and DCF at 10 μM when compared with a combination treatment of cisplatin and DCF at 10 μM. A similar result pattern was observed in treatment of 3MA together with cisplatin and DCF at 20 µM. Taken together, these results confirmed that DCF potentiated the cytotoxic effect of cisplatin in KATO/DDP cells via the activation of autophagy cell death.

### 2.3. DCF Potentiated Cytotoxicity of Cisplatin by Regenerating Intracellular ROS

Cisplatin exerts a cytotoxic effect, not only through the formation of platinum DNA adducts, but also via the generation of ROS. It has been pointed out that cisplatin-resistant cancer cells become highly adapted to drug-induced oxidative stress by upregulating their antioxidant systems. Therefore, intracellular ROS level in a combination treatment of cisplatin and DCF was evaluated by using 2′,7′- dichlorofluorescin diacetate (DCF-DA) assay. As shown in Figure 3A, when KATO/DDP cells were treated with cisplatin and DCF at 20 μM, a significant increase was observed in ROS generation when compared with treatments of cisplatin alone. Moreover, when N-acetyl cysteine (NAC) 2 mM was added to cisplatin and DCF at 10 μM, cell viability increased from 61.9 ± 0.4 to 74.6 ± 0.5% when compared with a combination treatment of cisplatin and DCF. A similar trend was observed in NAC treated with cisplatin and DCF at 20 μM (Figure 3B). An increase in the antioxidant response by activation of the Nrf2 transcription factor resulted in the enhanced resistance of cancer cells to certain chemotherapeutic agents including cisplatin. Inversely, the inhibition of Nrf2 rendered cancer cells more susceptible to the drug. To investigate whether DCF affected Nrf2 activity, the nucleus translocation of Nrf2 was then determined. A combination treatment of DCF at 10 and 20 μM with cisplatin significantly decreased the nuclear translocation of Nrf2 when compared with the control (Figure 3C,D). The resulting data collectively affirmed that DCF could potentiate the cytotoxic effect of cisplatin by regeneration of ROS via Nrf2 transcriptional activity.

### 2.4. DCF Downregulated Nrf2 Downstream Target Genes

Nrf2 was determined to be a critical regulator of antioxidant and xenobiotic detoxification genes including *HO-1*, *SOD*, *NQO1* and *AKRs*. Overexpression of these antioxidant response genes leads to attenuation ROS-mediated cell death as well as increased inactivation of anticancer drug, which could further promote cancer cell resistance to therapy. Therefore, we examined whether DCF modulated redox homeostasis in KATO/DDP cells was via the regulation of Nrf2-downstream target proteins. As is shown in Figure 4A,B, a combination of DCF and cisplatin significantly reduced the expression of HO-1 and SOD1. Moreover, our previous study had determined that xenobiotic metabolizing enzymes, AKR1C1 and 1C3, were involved in the development of cisplatin resistance in KATO/DDP cells via modulation of redox homeostasis [6]. The expression levels of AKR1C1 and 1C3 in KATO/DDP cells were significantly reduced in the combination treatment of cisplatin and DCF when compared with cisplatin alone (Figure 4C,D). Moreover, the effect of DCF on AKRs activity was investigated by employing the aldose reductase enzyme activity assay, which was used to monitor NADPH-dependent oxidoreductases activity. The AKRs activity assay indicated that DCF at 100 μM could inhibit its activity by up to 50%, as is shown in Figure 4E. These outcomes would indicate that DCF modulated redox homeostasis in KATO/DDP cells by decreasing the Nrf2 downstream targeted enzymes. Additionally, the survival analysis data obtained from the online cBioPortal were used to evaluate the prognostic significance of AKR1C1 and AKR1C3 mRNA expression levels in stomach adenocarcinoma tissue samples obtained from The Cancer Genomic Atlas (TCGA) Pan-Cancer Atlas data. As is shown in Figure 4F, patients who exhibited high levels of expression for both AKR1C1 and AKR1C3 mRNA (25 samples) were associated with significantly shorter overall survival rates when compared with those of patients who had not displayed high expressions of AKR1C1 and/or AKR1C3 (88 samples) with log-rank *p* = 0.0455.

### 2.5. DCF Potentiated Cisplatin-Induced Cytotoxicity in KATO/DDP Cells by Down Regulation of the Cell Survival Signaling Pathway

The effect of DCF on the signaling pathways that are involved in cisplatin resistance were investigated. As is shown in Figure 5A,B, a combination treatment of DCF at 10 and 20 μM and cisplatin reduced the phosphorylation levels of both the Erk1/2 and p38 MAPK signaling pathways, while the non-phosphorylation of Erk1/2 and p38 had no effect. On the other hand, the degree of the phosphorylated form of Akt was inhibited by DCF at 10 and 20 μM when compared with cisplatin, although there were no changes in the degree of non-phosphorylation of the Akt levels (Figure 5C,D). The activation of the MAPKs and PI3K/Akt signaling pathways induced the stimulation of NF-κB and the AP-1 transcriptional activity, which then mediated the upregulation of certain survival proteins such as cFLIP, cIAP2, Bcl-2, Bcl-xL and XIAP while favoring the survival of cisplatin-resistant cells. As is shown in Figure 5E, the nucleus translocation of c-Jun and NF-κB was downregulated after DCF at 10 and 20 μM was added to cisplatin in KATO/DDP cells when compared with cisplatin alone. Moreover, the level of STAT3 in the nucleus decreased when a combination treatment of DCF and cisplatin was administered. Additionally, cell proliferation and survival protein cyclin D1, cIAP2, Bcl-2 and Bcl-xL were downregulated in DCF at 10 and 20 μM in the cisplatin combination treatment of KATO/DDP cells when compared with the cisplatin treatment alone as described in Figure 5G,H. This would indicate that DCF could downregulate the relevant cell survival signaling pathways and lower the expression levels of cell survival proteins.

## 3. Discussion

DCF is also known as a non-selective Cox inhibitor and plays a role in the inhibition of prostaglandin synthesis, the inhibition of lipoxygenase enzymes and the activation of the nitric oxide–cGMP antinociceptive pathway [13]. Previous studies have reported that the use of NSAID in combination with chemotherapeutic drugs effectively potentiated anticancer efficacy [14,15]. Among these outcomes, DCF has been proven to be able to enhance the cytotoxicity of cisplatin in cancer cells [16]. Consequently, we explored the effect of DCF on cisplatin-induced KATO/DDP cell death. KATO/DDP cells showed 7.14 times greater resistance to KATOIII cells, which are understood to be suitable as a model for drug-resistant cells. Herein, we determined that DCF demonstrated a reduction in cell survival after being combined with cisplatin in KATO/DDP cells but not in KATOIII cells. This result is consistent with the findings of a previously published study conducted by Okamoto et al. [11,16], which indicated that DCF potentiated the antitumor property of cisplatin in a lung cancer mouse model, as well as in a cancer stem cell study.

Cisplatin damages tumors via several processes including apoptosis, ferroptosis and autophagy, depending upon the specifics of each type of cancer cell [17,18,19]. Cisplatin exerts cytotoxicity mainly by inducing apoptosis. Therefore, we tested whether the way in which DCF enhanced cisplatin-induced cytotoxicity in KATO/DDP cells could be attributed to cell apoptosis. However, a combination treatment of DCF and cisplatin revealed no significant differences in percent cell apoptosis when compared with treatments of cisplatin alone. Notably, many studies have shown that NSAIDs induced anticancer effects and increased chemotherapy sensitivity by regulating autophagy [20]. Celecoxib induced autophagy through inhibition of the PI3K/Akt signaling pathway and subsequently resulted in the inhibition of the proliferation among multidrug-resistant hepatoma cells [20]. The findings of our previous study indicated that the inhibitors of AKR1C1 and 1C3 activity enhanced cisplatin-induced KATO/DDP cell death by activating the autophagy pathway. Thus, the sensitizing effect of DCF on cisplatin-induced KATO/DDP cell death via the autophagy pathway was investigated. A combination of DCF and cisplatin raised the formation of the autophagy vacuoles and LCIIIB-2 in KATO/DDP cells when compared with treatments of cisplatin alone. To confirm that cell death occurred via the autophagy pathway, we inhibited autophagy by using 3MA, an autophagy inhibitor, and cell viability was then determined. Accordingly, 3MA reversed the effect of a combination treatment involving cisplatin and DCF by promoting cell survival to the same degree as cisplatin alone. Autophagy not only exhibits a pro-survival effect but also a pro-death mechanism that is dependent upon the type of chemotherapy being administered and the type of tumor tissue being treated. Previously published studies have reported that cisplatin exerts an antitumor effect by activating the autophagy cell death pathway in breast cancer subjects [21], which is consistent with our findings wherein DCF potentiated the cytotoxic effect of cisplatin via the autophagy cell death mechanism.

Apart from DNA damage, recent data suggested that cisplatin induces the ROS that is known to trigger cell death. Many recent studies have indicated that excessive ROS could lead to cell death via apoptosis and the autophagy pathway [22,23,24]. In the present study, an increase was observed in the regeneration of intracellular ROS after a combination treatment of DCF and cisplatin was applied to KATO/DDP cells. Conversely, inhibition of intracellular ROS by NAC increased cell survival to an equal level of cisplatin alone. This outcome is consistent with the findings of a study on oral squamous cell carcinoma, wherein cisplatin enhanced autophagy cell death by regulating intracellular ROS via the MAPKs signaling pathway [17]. This would indicate that a combination of DCF and cisplatin could induce autophagy-mediated KATO/DDP cell death by regenerating intracellular ROS. Remarkably, the Nrf2 transcription factor exerts the main antioxidant regulator that is involved in ROS detoxification by tightly regulating drug resistance during the treatment of tumors [25]. Therefore, inhibition of the Nrf2-dependent protective response should render cancer cells more susceptible to chemotherapeutic agents. With regard to exposure of cells to oxidative stress or chemotherapeutic drugs, the conformation of Keap1, a negative regulator of Nrf2, has changed resulting in ubiquitin ligase activity inhibition and an accumulation of Nrf2. Subsequently, Nrf2 translocated to the nucleus and activated the transcription of ARE-containing genes [26]. Herein, we found that a combination of DCF and cisplatin reduced Nrf2 nucleus translocation when compared with cisplatin. The resulting data confirmed that DCF could enhance cisplatin-induced intracellular ROS via modulation of Nrf2 activity.

Chemo-resistant cancer cells frequently express high levels of antioxidant enzymes, which in turn confer resistance to ROS-mediated cell death. The excessive activation of Nrf2 counterbalances the accumulated ROS by upregulating certain antioxidant enzymes including HO-1, NQO1 and SOD1 [4]. Furthermore, it has been reported that a reduction in antioxidant enzymes could sensitize cancer cells to cisplatin [27]. Our results indicated that there was a reduction in the expression of HO-1 and SOD1 after the addition of DCF to the cisplatin treatment when compared against treatment involving cisplatin alone. The upregulation of AKRs expression, phase I drug metabolism enzymes, by Nrf2 could therefore alter the cytotoxic response observed from the treatment with cisplatin. Overexpression of AKR1Cs has been identified as a mechanism of resistance to platinum drugs. Moreover, the results of our previous study indicated that cisplatin-resistant KATO/DDP cells exhibited an overexpression of AKR1C1 and 1C3, which play a role in developing chemoresistance via the regulation of redox homeosis [6]. Interestingly, survival analysis for incidences of stomach adenocarcinoma from TCGA data accessed by the cBioPortal indicated that patients who experienced an overexpression of AKR1C1 and 1C3 were associated with lower survival rates. This has also been reported in studies involving other types of cancer such as lung and liver cancer [28,29]. Therefore, we examined the effect of DCF on the expression of AKR1C1 and 1C3 in KATO/DDP cells. It was determined that a combination of cisplatin and DCF lowered the expression of AKR1C1 and 1C3. Additionally, DCF inhibited the NADPH-dependent oxidoreductases activity of AKRs. The outcomes of our study indicated that DCF played a role in the downregulation of the Nrf2 transcriptional factor and lowered the expression of antioxidant enzymes and AKR1Cs in KATO/DDP cells. This process in turn regenerated intracellular ROS and promoted cell death via the redox homeostasis-associated autophagy cell death mechanism.

Many signaling pathways and transcription factors have been observed to have an association with cisplatin resistance by promoting cell survival. The activation of MAPKs or the PI3K/Akt signaling pathway in resistance to cisplatin has been previously reported [30,31]. In the present study, DCF markedly reduced the phosphorylation and activation of Erk 1/2, p38 and Akt in the presence of cisplatin. This finding agreed with the outcomes of other previous studies that reported the inhibition of p38 and the Erk1/2 MAPKs pathway resulting in the upregulation of ROS and thereby sensitizing human tumor cells to cisplatin-induced cell death [32]. On the other hand, a combination of PI3K/Akt inhibitors and cisplatin have displayed a synergistic anti-tumor effect in chemo-resistant cancer cells including those of melanoma, breast and lung cancer [33,34,35,36]. Moreover, DCF induced leukemic cell death with increased intracellular ROS by inhibiting PI3K/Akt. [37]. Another study has found that activation of the PI3K/Akt/mTOR pathway suppressed autophagy cell death, whereas inhibition of this pathway induced autophagy [38].

Beyond Nrf2, other transcription factors, such as NF-κB, AP-1 and STAT-3, have been found to contribute to cisplatin resistance [39]. NF-κB and AP-1 regulated the expression of certain survival gene products including cIAP, XIAP, survivin, Bcl-2 and Bcl-xL [40], along with the proliferation of gene products, cyclin B1 and cyclin D1, as well as certain antioxidant enzymes that include SOD, glutathione S-transferase and HO-1 [41]. Furthermore, it has been reported that modulation of NF-κB and AP-1 activity could enhance the sensitivity of cancer cells to cisplatin [42,43]. Herein, the effects of DCF on the activity of NF-κB and AP-1 were investigated by observing the nucleus translocation of p65 and c-Jun in KATO/DDP cells. In this study, DCF reduced p65 and c-Jun translocation to the nucleus in cells treated with cisplatin. Presently, an abundant amount of evidence has indicated that STAT3, as well as cytokine IL6 and IL8, potentially form a positive feedback loop in cancer patients while enhancing cisplatin resistance [44,45]. In this study, we have demonstrated that DCF reduced STAT3 activity in KATO/DDP cells. Accordingly, we found that DCF downregulated NF-κB, AP-1 and STAT3 in a combination treatment with cisplatin. Thus, the effect of DCF on the expression of survival proteins was examined. Notably, a combination treatment of DCF and cisplatin reduced the expression of Bcl-2, Bcl-xL, cIAP and cyclinD1. Recent findings have revealed a novel function of anti-apoptotic proteins, such as Bcl-2 and Bcl-xL, as negative regulators of autophagy [46]. These results suggested that the way that DCF enhanced cisplatin-induced cell death was at least in part accomplished by downregulating cell survival proteins and the control of NF-κB, AP-1 and STAT3 activity.

In conclusion, DCF could enhance the cytotoxicity of cisplatin by increasing intracellular ROS, which further activated the autophagy cell death process. Moreover, DCF downregulated the MAPKs, Akt, NF-κB, AP-1 and STAT3 signaling pathways, which then lowered the level of expression among cell survival proteins with regard to the cisplatin resistance of the SRCGC cells line.

## 4. Material and Method

### 4.1. Chemical and Reagents

Dulbecco’s modified eagle medium (DMEM) and penicillin–streptomycin were supplied from Gibco (Grand Island, NY, USA), while fetal bovine serum (FBS) was purchased from Thermo Scientific Company (Waltham, MA, USA). Apoptosis annexin V and PI was obtained from Biolegand (San Diego, CA, USA.), whereas 3MA, MDC, DCF-DA, and cisplatin were obtained from Sigma-Aldrich (St. Louis, MO, USA). DCF and NAC were obtained from MedChemexpress (Princeton, NJ, USA). Aldose reductase inhibitor screening kit was purchased from Biovision (Waltham, MA, USA). KATO III cell lines was obtained from American Type Culture Collection (Manassas, VA, USA). Antibodies specific to AKR1C1, AKR1C3, PARP1, LC3B-II, Nrf2, HO-1, SOD1, p38, p-p38, ERK, p-ERK, NFκB, STAT3, c-Jun, Akt, p-Akt, cyclin D1, cIAP2, Bcl-2, Bcl-xL were purchased from Abclonal (Woburn, MA, USA).

### 4.2. Cell Culture Condition

Cisplatin-sensitive KATOIII and cisplatin-resistant KATO/DDP cell lines were cultured in DMEM containing 1% penicillin and streptomycin together with 10% FBS. The cells were maintained at 37 °C with 95% humidified air and 5% CO_2_. Acquired cisplatin-resistant KATO/DDP cell line was generated as previously mentioned in our work [6]. The resistance property of KATO/DDP cells was maintained by treating with cisplatin at 3 µM, for 72 h, and then the cells were cultured for at least two passages in drug-free media before each experiment.

### 4.3. Cytotoxicity Assay by Using Trypan Cell Exclusion

To identify the cytotoxicity, gastric cancer cells KATOIII and KATO/DDP at a concentration of (1.2 × 10^5^) were plated in 12-well plate, then treated with 0–100 µM concentration of cisplatin alone, cisplatin together with DCF, cisplatin in combination with DCF and 3MA or cisplatin in combination with DCF and NAC, at 37 °C, for 48 h. After treating with desired drugs for 48 h, the cells were collected, and resuspended in 1 mL of DMEM. Then, 100 µL of cell suspension was taken and stained with 10 µL of 0.4% *w*/*v* trypan blue dye. The cell viability was assessed by detecting the trypan blue staining viable cells under microscope. Percentage of viable cells was calculated compared to that of control group.

### 4.4. Apoptosis Assay

KATO/DDP cells at a concentration of (2 × 10^5^ cells) were treated with or without cisplatin alone and cisplatin together with DCF and incubated for 48 h, at 37 °C. After 48 h, the cells were harvested and washed with phosphate-buffered saline (PBS). The collected cells were then stained with annexin V and PI and incubated for 30 min, at room temperature, in dark. After 30 min, the stained cells were analyzed by using the BD FACScanTM flow cytometer (BD Biosciences, San Jose, CA, USA). The results were analyzed by using CytoExpert software.

### 4.5. Monodensyl Cadaverine Staining

KATO/DDP cells at a concentration of (2 × 10^5^ cells) were treated with cisplatin alone or cisplatin together with DCF in 12-well plates and incubated for 24 h, at 37 °C. Then, the cells were washed with ice cold PBS, resuspended in incomplete DMEM, and stained with MDC dye for 30 min, at 37 °C. The cells were then washed with PBS twice to remove the excess MDC dye. They were then visualized with a fluorescence microscopy at an excitation wavelength of 460–500 nm and emission wavelength of 512–542 nm.

### 4.6. 2′,7′-Dichlorofluorescin Diacetate Assay

Intracellular ROS was measured using DCF-DA assay. KATO/DDP at 4 × 10^5^ cells were treated in 12-well plates with either cisplatin alone or cisplatin with DCF for 4 h. The treated cells were harvested and washed with ice cold PBS, and then stained with DCF-DA at 5 µM for 30 min at 37 °C. The cells were then washed with PBS and lysed with 90% (*v*/*v*) dimethyl sulfoxide in PBS. The fluorescence intensity of the lysed cells was measured at an excitation and emission wavelength of 485/525 nm by a fluorescent plate reader.

### 4.7. Western Blotting Analysis

KATO/DDP at 5 × 10^5^ cells were treated with cisplatin alone or cisplatin combined with DCF at 5 µM, 10 µM and 20 µM for 24 h. Then, the cells were collected and washed with PBS and centrifuged at 9279× *g* for 1 min, at 4 °C. The cell pallets were lysed for 20 min on ice in RIPA lysis buffer containing protease inhibitors (1 mM PMSF, 10 μg/mL aprotinin, 10 μg/mL leupeptin), then centrifuged at 13,362× *g* for 15 min, at 4 °C, to remove insoluble matters. The supernatant was collected, and protein concentration was determined by Bradford assay. Total protein at 20 mg extracted from samples were loaded and separated into SDS polyacrylamide gel electrophoresis at 100 V for 1 h, before being transferred to nitrocellulose membrane. After 1 h incubation with 5% skimmed milk, the nitrocellulose membrane was then washed and incubated with desired antibody for overnight, at 4 °C. The secondary antibodies at a concentration of 1:20,000 were used to incubate the membrane for 2 h, at room temperature. The membranes were visualized by using chemiluminescent, which were then exposed to X-ray film. Quantitative expression of each protein was determined by analyzing band density by using TotalLab TL120.

### 4.8. Aldose Reductase Inhibitor Screening Assay

DCF at a concentration of 0–500 µM were prepared in PBS and is tested for aldose reductase inhibition activity by using aldose reductase inhibitor screening kit according to manufacturer’s instruction. The aldose reductase activity was monitored by reduction in reading of absorbance at 340 nm by spectrophotometer.

### 4.9. Bioinformatics Analysis

A total of 168 stomach adenocarcinoma patients from TCGA PanCancer atlas database was subjected to analyze the overall survival rate by using an open assess tool cBioPortal. The high expression group represents the patient who has mRNA expression z-scores relative to all samples (log RNA Seq V2 RSEM) greater than 0.5. The overall survival rate of the group of patients who exhibited high level of expression for both AKR1C1 and AKR1C3 mRNA was compared with the group of patients who had not displayed high expression of AKR1C1 and/or AKR1C3 by using Logrank Test *p*-value.

### 4.10. Statistical Analysis

Each experiment was performed for three independent experiments. Statistical analysis was conducted by using SPSS (version 22). All the data are expressed in mean ± SD. For the comparisons among multiple variables, we used one-way ANOVA in this study. The statistical significance level was considered as significant when *p* value was <0.05 and <0.01.

## Figures and Tables

**Figure 1 ijms-23-12066-f001:**
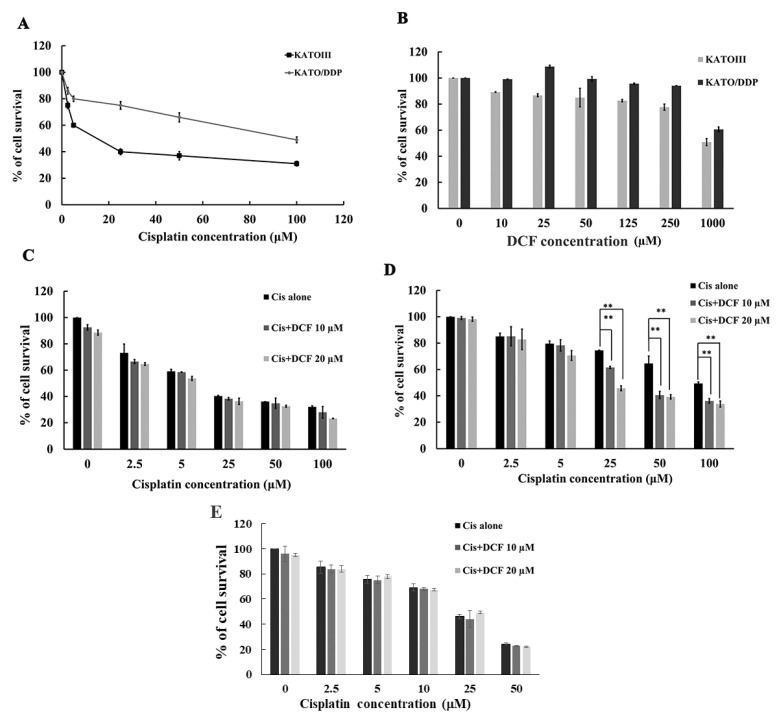
Effect of DCF on cisplatin cytotoxicity in SRCGC cells. (**A**) KATOIII and KATO/DDP cells were treated with cisplatin 0–100 µM concentration and incubated for 48 h, at 37 °C. Cell viability was assessed by using trypan blue cell exclusion assay. (**B**) KATOIII and KATO/DDP were treated with different concentration of DCF (0–1000 µM) for 48 h, and cell viability was assessed. KATOIII (**C**) and KATO/DDP (**D**) cells were treated with cisplatin (0–100 µM) either alone or in combination with DCF at 10 or 20 µM and incubated for 48 h, at 37 °C, and cell viability was assessed by using trypan blue cell exclusion assay. (**E**) Human dermal fibroblast cells were treated with cisplatin (0–50 μM) either alone or in combination with DCF at 10 and 20 μM and incubated for 48 h, at 37 °C, and the cell viability was determined by using MTT assay. Data were representative of three independent experiment as mean ± SD. ** represented *p* < 0.01. DCF = Diclofenac, Cis = Cisplatin.

**Figure 2 ijms-23-12066-f002:**
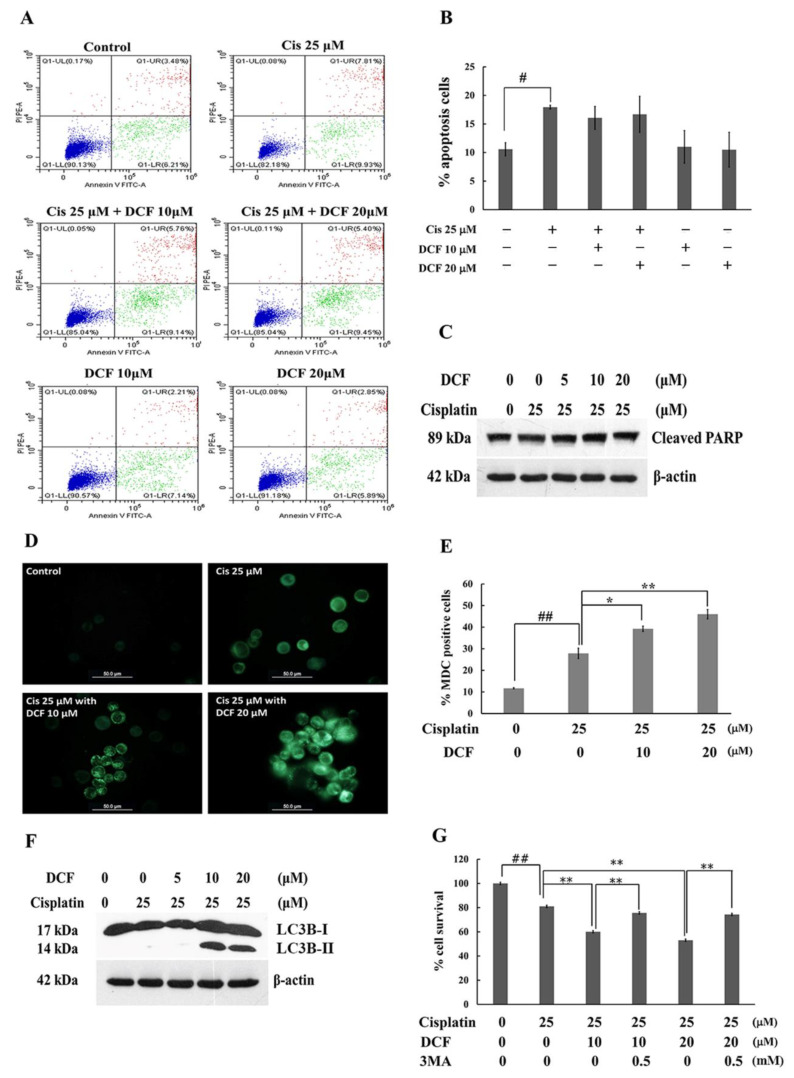
DCF in combination with cisplatin induced KATO/DDP cells death via autophagy pathway. The KATO/DDP cells were treated with cisplatin alone or cisplatin in combination with DCF at 10 or 20 µM for 24 h, at 37 °C. The apoptosis cells were detected by using annexin V and PI staining and analyzed by flow cytometer (**A**). Graphical representation of percentage of apoptosis cells (**B**). The combination effect of DCF and cisplatin on the expression of cleaved PARP in KATO/DDP cells was examined by Western blot analysis (**C**). Autophagy vacuoles formation was detected by labelling with MDC dye and detected under fluorescent microscope (**D**) and percentage of autophagy vacuoles was quantified and represent in bar graph (**E**). KATO/DDP cells were treated with cisplatin 25 µM either alone or combination with DCF 5, 10 or 20 µM and incubated for 24 h, and LC3B-II formation was detected by Western blot analysis (**F**). Cell viability of KATO/DDP cells treated with cisplatin 25 µM either alone or combination with diclofenac 10 or 20 µM, with or without 3MA after 48 h of treatment (**G**). Data were representative of three independent experiment as mean ± SD. ^#^ represented *p* < 0.05 and ^##^ represented *p* < 0.01 as compared to the control group, while * represented *p* < 0.05 and ** represented *p* < 0.01 as compared to cisplatin alone group. DCF = Diclofenac, Cis = Cisplatin, 3MA = 3-methyladenine.

**Figure 3 ijms-23-12066-f003:**
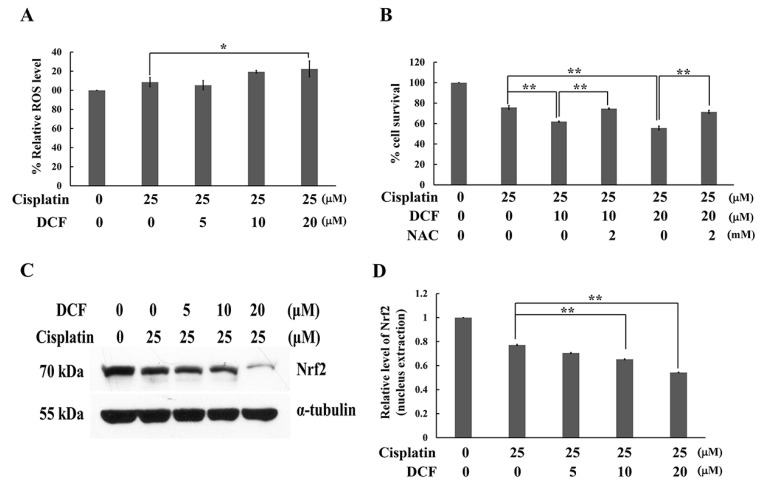
DCF potentiated the cisplatin cytotoxicity in KATO/DDP cells via the regeneration of intracellular ROS. (**A**) KATO/DDP cells were treated with either cisplatin alone or in combination with DCF and intracellular ROS was detected at 4 h after treatment by using DCF-DA dye, then the fluorescent intensity was determined. (**B**) KATO/DDP cells were treated with either cisplatin alone or in combination with DCF 10 or 20 μM with or without NAC 2 mM and cell viability was assessed. (**C**) KATO/DDP cells were treated with either cisplatin alone or in combination with DCF 5, 10 or 20 μM for 24 h, and the expression level of Nrf2 in the nucleus were detected by Western blot analysis. (**D**) Quantitative representation of band density for Nrf2 (nuclear extraction). Data were presented as mean ± SD. * Represented *p* < 0.05, while ** represented *p* < 0.01. DCF = Diclofenac, NAC = N-acetyl cysteine, ROS = Reactive oxygen species.

**Figure 4 ijms-23-12066-f004:**
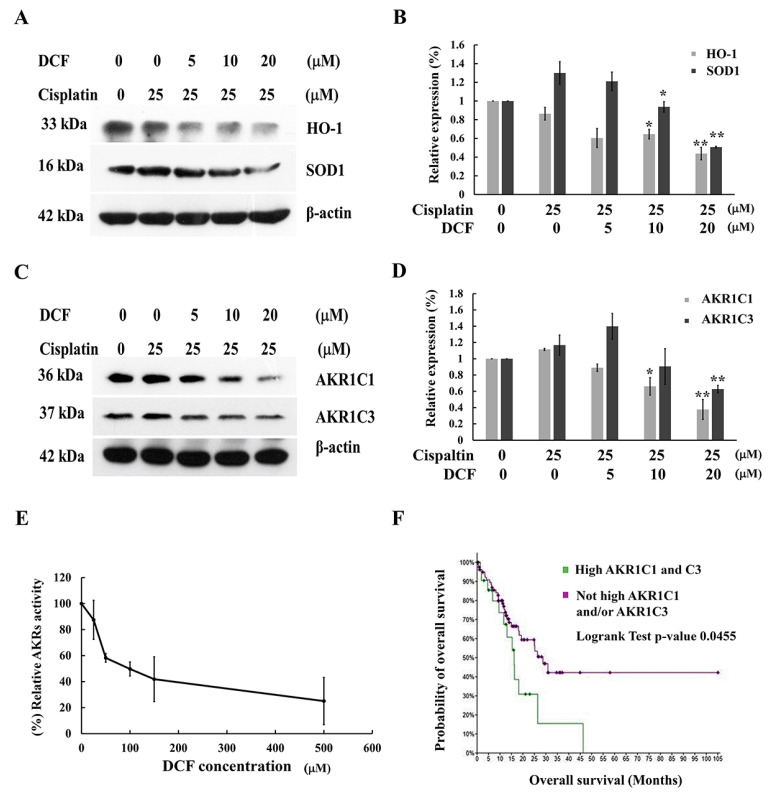
DCF modulated Nrf2 downstream target genes in cisplatin-treated KATO/DDP cells. (**A**) KATO/DDP cells were treated with cisplatin alone or combined with DCF 5, 10 or 20 μM for 24 h, and the expression of HO-1 and SOD1 were determined by Western blot analysis. (**B**) Quantitative representation of band density for HO-1 and SOD1. (**C**) The expression of AKR1C1 and 1C3 were evaluated by Western blot analysis, and (**D**) densitometric and statistical analysis of protein quantification data are presented as histogram. (**E**) The effect of DCF on AKRs activity was measured by using aldose reductase activity kit. (**F**) The overall survival curve comparing the gastric cancer patients between high AKR1C1 and 1C3 mRNA expression and not high AKR1C1 and/or 1C3 mRNA expression by using cBioPortal for cancer genomics. All experiments were performed at least three times. * Represented *p* < 0.05, while ** represented *p* < 0.01 as compared to the cisplatin alone. DCF = Diclofenac, AKRs = Aldoketoreductases.

**Figure 5 ijms-23-12066-f005:**
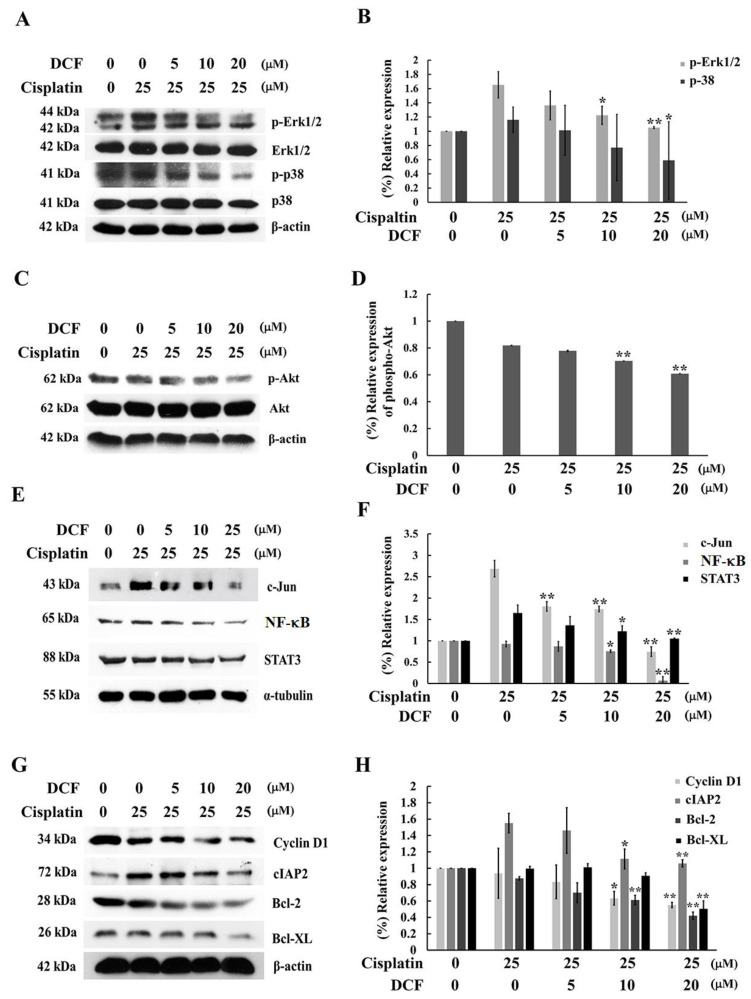
DCF downregulated survival proteins and modulated survival signaling pathways in KATO/DDP cells. The cells were treated with either cisplatin alone or in combination with DCF at 5, 10 or 20 μM. After incubation for 24 h, the whole cell lysate was prepared and subjected to investigate the activation of Erk1/2 and p38 MAPKs (**A**) and Akt (**C**) signaling molecules by Western blot analysis. Densitometric and statistical analysis were performed to determine the relative phosphorylation of Erk1/2, p38 (**B**) and Akt (**D**). The nucleus extraction was used to analyze the transcriptional activity of c-Jun, NF-κB and STAT3 (**E**) and histogram graph represented the band intensity of c-Jun, NF- κB and STAT3 (**F**). The expression levels of survival and proliferation proteins including cyclinD1, cIAP2, Bcl-2 and Bcl-XL were examined by Western blot analysis (**G**), and the histogram represented the band intensity of survival proteins (**H**). All experiments were performed at least three times. * Represented *p* < 0.05, while ** represented *p* < 0.01 as compared to the cisplatin alone. DCF = Diclofenac.

## Data Availability

The total of 168 stomach adenocarcinoma patients from TCGA PanCancer atlas database was subjected to analyze the overall survival rate was found in https://www.cbioportal.org/study?id=6342f3dd08af4268968f0d72, accessed on 7 September 2022.

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
