# Peer review of "Diclofenac Sensitizes Signet Ring Cell Gastric Carcinoma Cells to Cisplatin by Activating Autophagy and Inhibition of Survival Signal Pathways"

_ijms, 2022, doi:10.3390/ijms232012066_

Round 1

Reviewer 1 Report

ijms-1933506

Title: Diclofenac Sensitizes Signet Ring Cell Gastric Carcinoma Cells to Cisplatin by Activating Autophagy and Inhibition of Survival Signal Pathways

Corresponding author: Supachai Yodkeeree

In general, the data in present studies are good and support the major conclusions of this manuscript. However, following issues need to be considered prior to considering the manuscript of publication.

1.    According to the PubMed investigation, several research papers have been confirmed that diclofenac promotes the death of cells by inducing autophagy in several human cancer cells. Therefore, it is advisable to add the results of these previous studies to the introduction part.

2.    Abbreviations: The use of abbreviations when writing a paper has many advantages besides simplicity of expression. To use an abbreviation, first write the abbreviation in parentheses after the full name, and then use the abbreviation from Introduction to the final conclusion. In this paper, in most cases, the abbreviation is written first and then the full name is written in parentheses. Only in abstract and figure legend do it separately. In particular, in review papers, there is no choice but to use a lot of abbreviations. In this case, it is necessary to define and use abbreviations very systematically.

3.    Reference number 1: Reference on gastric cancer-related statistics is too old, replace it with the latest reference(s).

4.    Materials and Methods section - When naming a particular chemical company, you must provide location information such as company name, city and/or state (abbreviation in the USA and Canada) and country. Once you have named a company with the information, you should only mention a company’s name thereafter. Sigma should be Sigma-Aldrich.

5.    Line 33: Define H. pylori.

6.    Figure legend for Figure 1 and more: If an abbreviation is used in the figure, write the full name in figure legends.

7.    Line 93 and more: The unit display for temperature is strange, so please check again throughout the manuscript.

8.    Line 98: There is an unnecessary space between ‘cells’ and ‘death’, so fix it.

9.    Line 124 vs. Line 386 for 60 % and 1%: When using % after a number, either a space must be placed after the number or it must be used after the number. Usually, % is appended to a number.

10.    Lines 427 and 429: The speed of the centrifuge should be expressed in gravity rather than rpm. Example: 12,000 rpm.

11.    English: The manuscript has numerous grammatical and typographical errors that should be corrected. Even though the authors submitted certificate of English proofreading, but proofreading by an experienced scientist is also required. The authors capitalize the first letter of the chemical name without needing to use it, so change all lowercase letters. Examples: Line 375, Line 376, Line 377, etc.

12.    Lines 140 and 397: Leave a space between the number and the unit.

13.    Proof readings and corrections by very experienced scientists around the authors are necessary.

14.    Reference section: Author should consult and peruse carefully recent issues of the journal, International Journal of Molecular Sciences, for format and style. The first letter of the title must be in upper case, and the rest must be in lower case. Examples: 2, 3, 5, 7, 17, 18, etc.

Overall, the manuscript can be considered to publication after major revision as indicated above.

Reviewer 2 Report

I would like to congratulate the idea of combining a Cox inhibitor with cisplatin. I was interested in article and the results from the study. I think it was planned logically and the manuscript is really well written. It is worth publication in IJMS.

First of all, a combination treatment of DCF and cisplatin revealed no significant differences in percent cell apoptosis when compared with treatments of cisplatin alone. Authors checked the effect on autophagy and it was proved that concomitant treatment of DCF and cisplatin significantly enhanced autophagic cell death due to overproduction of intracellular reactive oxygen species (ROS). Such a combination also inhibited the expression of survival proteins including Bcl-2, Bcl-xL, cIAP1 and cyclin D1 in 22 KATO/DDP cells when compared with cisplatin alone. This was due, at least in part, to reduced MAPKs, Akt, NF-kappaB, AP-1 and STAT-3 activation.  The results are promising and further studies should be done in in vivo model.

I would like to know the effect of such a combination in normal cells  (human skin fibroblasts or other)? Is it less cytotoxic than cisplatin alone?

Round 2

Reviewer 1 Report

 Accept in present form.